# Investigating Various Products of IMERG for Precipitation Retrieval over Surfaces with and without Snow and Ice Cover

Alireza Arabzadeh [1] and Ali Behrangi [1,2,*]

1   Department of Hydrology and Atmospheric Sciences, University of Arizona, Tucson, AZ 85721, USA; arabzadeh@email.arizona.edu
2   Department of Geosciences, University of Arizona, Tucson, AZ 85721, USA
*   Correspondence: behrangi@email.arizona.edu

**Abstract:** Precipitation rate from various products of the integrated multisatellite retrievals for GPM (IMERG) and passive microwave (PMW) sensors are assessed with respect to near-surface wet-bulb temperature ($T_w$), precipitation intensity, and surface type (i.e., with and without snow and ice on the surface) over the contiguous United States (CONUS) and using ground radar product as reference precipitation. IMERG products include precipitation estimates from infrared (IR), combined PMW, and combination of PMW and IR. It was found that precipitation estimates from PMW products generally have higher skills than IR over snow- and ice-free surfaces. Over snow- and ice-covered surfaces: (1) most PMW products show higher correlation coefficients than IR, (2) at cold temperatures (e.g., $T_w < -10\,^{\circ}C$), PMW products tend to underestimate and IR product shows large overestimations, and (3) PMW sensors show higher overall skill in detecting precipitation occurrence, but not necessarily at very cold $T_w$. The results suggest that the current approach of IMERG (i.e., replacing PMW with IR precipitation estimates over snow- and ice-surfaces) may need to be revised.

**Keywords:** rainfall; stage IV; infrared; passive microwave; surface type; wet-bulb temperature

## 1. Introduction

Accurate precipitation estimation with high spatiotemporal resolution is key to many hydrologic studies. Rain gauges and ground radars have enabled high-quality observation and estimation of precipitation at a point or at regional scale and satellite observations have enabled precipitation estimates with global coverage at subdaily temporal sampling, which is important for hydrologic applications [1]. Advancing the global precipitation estimate is critical to better understand the current state of Earth's climate and future changes [2,3] and to help society through its various applications [3]. Accordingly, efforts have been devoted to evaluate satellite precipitation products over different regions to determine their errors and identify areas for future improvements [4–8].

The high temporal sampling from satellite observations comes through the use of infrared images from geostationary satellites, multiple PMW sensors on low-earth-orbiting satellites, or a combination of both. PMW sensors often provide more information about the hydrometeors, thus tend to result in more accurate precipitation estimates than precipitation retrieval based on IR data. However, PMW-based precipitation estimation may also face large uncertainties due to several factors including errors related to the poor understanding of precipitation microphysics, difficulties in distinguishing between light rain and cloud [9–11], and challenges in determining surface emissivity, especially over snow and ice [12,13].

The methods for blending IR- and PMW-based precipitation estimates have been different across different merged precipitation products. Precipitation estimation from remotely sensed information using artificial neural networks (PERSIANN) [14] and PERSIANN cloud classification system (PERSIANN-CCS) [15] are mainly based on IR brightness

temperature-precipitation rate relationships established using geostationary IR observations and PMW precipitation estimates as reference precipitation. Similar to most IR based methods, grids with colder cloud-top brightness temperatures are assigned to produce higher precipitation intensity. PERSIANN-CCS is different from PERSIANN because the IR-precipitation relationship is determined for different cloud classes, each determined based on features of groups of connected-grids called clusters [7]. On the other hand, products such as the integrated multisatellite retrievals for GPM (IMERG) [16], Climate Prediction Center's morphing technique (CMORPH) [17], and JAXA's global satellite mapping of precipitation (GSMaP) [18] use PMW precipitation as their main input and IR precipitation might be used to fill gaps in time. For example, if the time interval from the nearest PMW observation is longer than 30 min, IMERG tends to use a combination of IR and PMW estimates and, for time intervals of more than 90 min, the precipitation rate mainly comes from IR-estimate. This is because by about $\pm$ 90 min the IR precipitation has shown a higher correlation with reference precipitation than the propagated microwave precipitation estimates [16].

IMERG also uses IR precipitation estimate over snow and ice surfaces, regardless of the time distance from PMW observations and type of PMW sensors [16]. This is because PMW precipitation estimates are perceived as unreliable over snow and ice surfaces. However, the performance of IR precipitation over snow and ice surfaces has also not been well investigated. Furthermore, it has been found that different PMW sensors have dissimilar skills for precipitation retrieval over land and ocean [19]. It is important to assess how precipitation estimates from different PMW sensors (or a combination of them as used in IMERG) are compared with IR precipitation over surfaces with and without snow and ice. IMERG prefers precipitation estimates from PMW imagers over PMW sounders, which may not always be the best choice over land [19]. Examples of recent PMW imagers used in IMERG are the special sensor microwave imager/sounder (SSMIS) on the Defense Meteorological Satellite Program (DMSP) platforms, the GPM microwave imager (GMI) on the GPM Core Observatory satellite, and the advanced microwave scanning radiometer earth observing system (AMSR-E) on the Aqua satellite; and its follow-on satellite (AMSR2) on board the GCOM-W1 satellite. Among the main PMW sounders used in IMERG are the advanced technology microwave sounder (ATMS) on board Suomi National Polar-Orbiting Partnership (Suomi-NPP) and NOAA-20 satellites; the microwave humidity sounder (MHS) on board NOAA-18, NOAA-19, MetOp-A, MetOp-B, and MetOp-C satellites. In IMERG, IR precipitation is obtained from PERSIANN-CCS.

Besides considering surface conditions (e.g., surfaces with and without snow and ice) in comparison of PMW and IR precipitation, it is important to perform the analysis as a function of surface temperature, as surface temperature is used to discriminate the precipitation phase. Behrangi et al. [20] compared AMSR-E and CloudSat precipitation detection as a function of surface air temperature at 2 m (T2m) and showed that AMSR-E significantly underestimates CloudSat precipitation for T2m below the freezing temperature. Zhang et al. [21] also used T2m to evaluate high-resolution ($0.1°$/hourly) precipitation estimates from the weather research and forecasting (WRF) model and IMERG over the central United States. Results show that the WRF estimates exhibit higher correlations with the reference data when the temperature falls below 280 K, while IMERG estimates show higher correlation than WRF for T2m greater than 280 K. They also showed that the complementary behavior of the WRF and the IMERG products conditioned on T2m does not vary with either season or location. However, their study did not provide information on the performance of PMW and IR precipitation estimates (that are used in IMERG) and did not consider surface condition (i.e., surfaces with and without snow and ice) that determines the use of IR or PMW precipitation components in IMERG.

In the present study, we focus on evaluating the performance of IMERG and its precipitation components over surfaces with and without sea and ice cover and as a function of Tw. In addition, analysis as a function of precipitation rate provides insights into the performance of the precipitation products under light, moderate, and intense

precipitation events. This investigation adds insights to refine strategies for combining IR and PMW precipitation estimates in the merged products such as IMERG. Therefore, this study is different from previous studies in which the final products of IMERG is assessed using ground reference [5–7,22]. Results of this study can improve future versions of satellite-based precipitation products and provide insight for new sensors' design and their performance over different conditions and regions.

## 2. Materials and Methods

### 2.1. Comparison Approach and Metrics

Using three years of stage IV data (2015–2017) as a reference over the CONUS, the performance of each product is assessed within each one-degree $T_w$ bin, separately over surfaces with and without snow and ice cover. Precipitation thresholds of 0, 0.1, 0.3, 1, and 2 mm/h are used to separate precipitation from nonprecipitation. For example, by using a threshold of 2 mm/h, only grids with precipitation rates of greater than 2 mm/h are considered as precipitating and anything less than that is considered as nonprecipitating, so the emphasis will be more on intense precipitation events. The calculation of $T_w$ is described in [23] and uses 2 m air temperature, 2 m dewpoint temperature, and surface pressure that are obtained from reanalysis. The score metrics used in this study are: probability of detection (POD), false alarm ratio (FAR), bias, and Heidke skill score (HSS) for assessment of precipitation occurrence as well as correlation coefficient (CC) and volume bias (VBias) for assessment of precipitation rate. HSS provides a more generalized skill score for assessing the accuracy of the predictions relative to the random chance. In other words, HSS shows the fraction of correct predictions by excluding correct predictions due to random chance. The range of the HSS is $-\infty$ to 1. Negative values indicate that the forecast by chance is better, 0 means no skill, and 1 means a perfect forecast. The ideal score for Bias and VBias is 1. Bias is calculated by dividing the number of estimated precipitation occurrences from each product by the corresponding value from the reference product (here Stage IV). VBias is similar to bias, but is calculated by dividing the amount of estimated precipitation by the corresponding value from observation. It should be noted that bias of 1 alone does not necessarily indicate a perfect prediction, because bias of 1 means that the number of grids identified as precipitation is the same for the two products being compared. However, a product might miss of falsely determine precipitation occurrence as can be inferred from POD and FAR. Details for calculation of the above metrics are provided in [24].

Note that it is known that most satellite products underestimate orographic precipitation enhancement and are not able to capture that accurately [25]. Therefore, to separate this effect from our analysis, using maps of mountains (see Section 2.2), regions susceptible to orographic precipitation enhancement were removed from the analysis.

### 2.2. Dataset

A brief description of the products used in this study is provided below:

- IMERG Products

The latest version of the IMERG products (V06) is used in this study. IMERG provides gridded precipitation maps with high spatiotemporal resolution (0.1 × 0.1 deg. every $\frac{1}{2}$ h) within 90° S-N by blending precipitation estimates from two sources: (1) IR images using PERSIANN-CCS, and (2) GPM microwave imager (GMI) and a constellation of GPM PMW sensors. IMERG uses the latest version of PMW precipitation products (V05) retrieval based on the 2017 version of the Goddard profiling algorithm (GPROF2017) [26]. Details of the IMERG algorithm are described in Huffman, Bolvin [27] and in brief includes four main steps: (1) precipitation estimates from the GPM constellation radiometers are gridded, intercalibrated to the radar-microwave combined product (2BCMB), and combined into half-hourly 0.1° × 0.1° fields (variable name: "HQprecipitaiton"), hereafter referred to as IMERG-HQ, (2) maps of half-hourly IR precipitation rate ("IRprecipitation") are calculated using PERSIANN-CCS, hereafter referred to as IMERG-IR, (3) MW and IR estimates

are used to create half-hourly estimates ("precipitationUncal") by utilizing the Climate Prediction Center morphing Kalman filter (CMORPH-KF) Lagrangian time interpolation scheme, and (4) the multisatellite half-hour estimates are adjusted so that they sum to a monthly satellite-gauge combination ("precipitationCal"). This bias-adjusted product is referred to as IMERG-Final which is available ~3.5 months after observation for accurate estimation with monthly gauge adjustments to reduce bias. IMERG also reports sources of PMW sensors "HQprecipSource", but it does not specify the platform names. Note that IMERG provides users with two other runs. The early run is available ~4 h after observation for real-time applications such as flood prediction and includes only forward morphing and may not benefit from all PMW sensors. The late run (IMERG-Late) is available ~14 h after observation, implementing forward and backward morphing. The current version of IMERG-Late (V6) does not apply climatology bias adjustment, but the future versions will.

In this study, we use IMERG-Late, IMERG-Final, IMERG-HQ (PMW-only), and IMERG-IR (IR-only) products available at half-hour at 0.1 × 0.1 degree resolution (i.e., about 11 km × 11 km at equator). As discussed in Section 1, in IMERG, IR is used over snow and ice surfaces or when the time distance from PMW observations is large. For the period of this study, IMERG-HQ uses imagers (AMSR-2 and GMI), sounders (ATMS, and MHS), and combined imager and sounder (SSMIS) sensors. All of these sensors, regardless of their platform, are used in our analysis. When there are available data from sounders and imagers, IMERG prioritizes imagers over sounders with consideration of observation time closest to the center of the half-hour (Huffman et al., 2020).

- National Centers for Environment Prediction (NCEP) Stage IV

The hourly Stage IV product is used in this study as reference data to analyze IMERG products' performance over CONUS. Since 2002, the hourly Stage IV product provides hourly estimates using Z-R relationship from over 150 Doppler Next Generation Weather Radars (NEXRAD), and a combination of 5500 hourly rain gauge measurements to produce hourly 4 × 4 km resolution data [28,29]. As of April 2017, this dataset includes Alaska and Puerto Rico stations. Stage IV benefits from manual quality control (QC) on stage III data gathered at each river forecast center (RFC) unlike stage II, which does not include manual QC (https://data.eol.ucar.edu/dataset/21.093 (accessed on 15 February 2021)). Including quality control, gauge measurements give assurance of the quality of the data. This becomes more important in the case of snowfall measurements when the temperature is below the freezing point because most of the radar-based estimations under freezing conditions are controversial [30]. Cocks and Martinaitis [31] shows the value of nine winter precipitation events over the Rocky Mountains is in good agreement with gauge measurements. Altogether we found out NCEP stage IV data are the most suitable and convenient for our study with fairly reliable accuracy [30].

- ERA5-Land

This dataset is a replay of ECMWF original land component ERA5 climate reanalysis with a finer resolution at ~9 km grid spacing with an hourly time interval. This dataset provides surface variable data from 1981 to 2–3 months before the present. In this study to calculate wet-bulb temperature, three different variables were obtained from ERA5-Land including 2 m air temperature, 2 m dewpoint temperature, and surface pressure for 2015 to 2017 over CONUS. We used wet-bulb temperature to distinguish rainfall from snowfall following as it is a better separator than air temperature [23,32,33].

- NOAA Autosnow Product

The NOAA autosnow product provides daily surface ice and snow map with global coverage. It is a gridded product with 0.04° lat/lon resolution. It uses data gathered from different sensors on various satellites. For details of sensors used in this product see Romanov [34]. IMERG uses autosnow in two stages: (1) mask snow- and ice-covered surfaces for Kalman statistics computation and, (2) mask IMERG precipitation estimates. In this study, we use autosnow to delineate surfaces with and without snow and ice cover.

- K3 Mountain Map

In this study to detach the issues related to the underestimation of orographic precipitation by satellite products from our analysis, K3 mountain mask was used. K3 provides a GIS-based, global map of mountains derived from digital elevation models (DEM) with a 250 m resolution. It has been developed after K1 and K2 mountains raster with a coarser resolution [35–37]. For comparison of different K mountain maps see Roger and Charlie [38]. K3 characterizes mountains into four different groups: high and high-scattered mountains with elevation exceeding 900 m and low and low-scattered mountains with elevation ranging between 301–900 m. K3 maps are found useful to identify regions that might contain orographic precipitation (personal communication with Dr. Paula Brow of Colorado State University). Since in this study we are not focusing on orographic precipitation and radar precipitation estimates over mountainous regions may not be accurate [39], we removed all high and high-scattered mountains (see Supplementary Figure S1) from our analysis.

## 3. Results

The results of this study are presented under three main sections: (1) general characteristics and differences of IMERG and stage IV products including spatial distribution and seasonality accompanied by a case study, (2) analysis of IMERG components versus stage IV data over snow-and ice-covered and snow- and ice-free surfaces with different intensities and, (3) investigating the performance of individual PMW sensor types.

### 3.1. General Characterization

Figure 1 shows seasonal mean precipitation maps from stage IV (Figure 1a–d), IMERG-Final (Figure 1e–h), IMERG-HQ (Figure 1i–l), IMERG-IR (Figure 1m–p), and IMERG-Late (Figure 1q–t) using three years (2015–2017) of data over CONUS, and annual averages of these products are shown in Figure S2. For a more detailed assessment, maps of seasonal differences in mean precipitation rate between IMERG products and stage IV are also plotted in Figure 2. From these two figures, few points can be highlighted: (1) because IMERG-Final is bias-adjusted with gauges at monthly scale, it is closest to stage IV in terms of both magnitude and pattern, (2) over mountainous regions, mainly in the west, IMERG products tend to underestimate precipitation rates during winter (DJF) where snow and ice are on the surface. This is also observed in spring (MAM) and fall (SON), but not in summer (JJA), although the underestimation is more noticeable for IMERG-HQ than IMERG-IR that is consistent with previous studies [40,41], (3) IMERG products (except IMERG-Final, which is bias-adjusted) show larger precipitation amount than stage IV over the central and eastern parts of the CONUS, especially during DJF and MAM. It is not clear if this overestimation, especially during the DJF, is due to overestimation of rainfall, snowfall, or due to surface conditions (e.g., snow and ice on the surface). Therefore, to better understand the performance of the satellite products, a more detailed analysis as a function of surface type and precipitation phase is needed.

The observed underestimation over the mountainous west can be due to poor skill of satellite products in capturing orographic enhancement [25,42] and the fact that most of the annual precipitation over this region occurs in wintertime [43] and a large fraction of the that is through the atmospheric rivers [44,45]. However, it is important to note that radar beam blockage may also contribute to the lower quality of stage IV product in mountainous regions. Most of these regions are masked out using the K3 mask applied in this study.

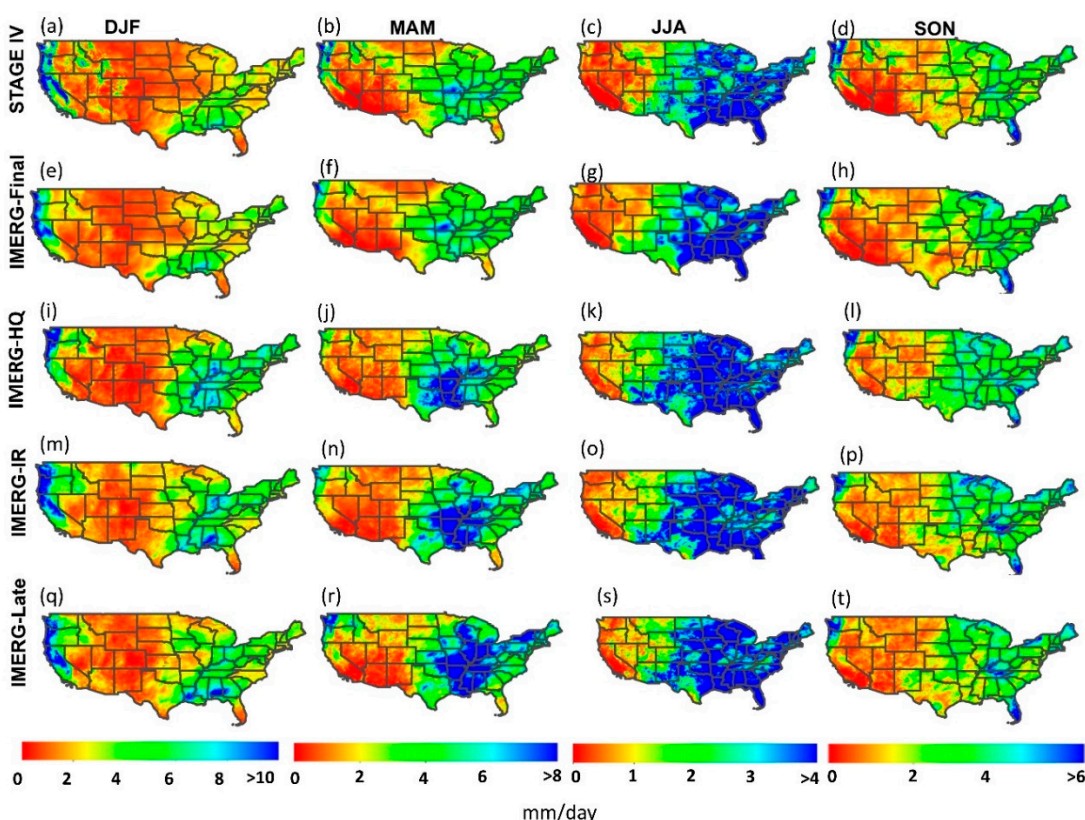

**Figure 1.** Spatial distribution of mean seasonal precipitation rate (mm/day) over the CONUS using stage IV (**a**–**d**) and various IMERG products (**e**–**t**), each presented in a row. Average intensity is calculated from three years (2015–2017) of data.

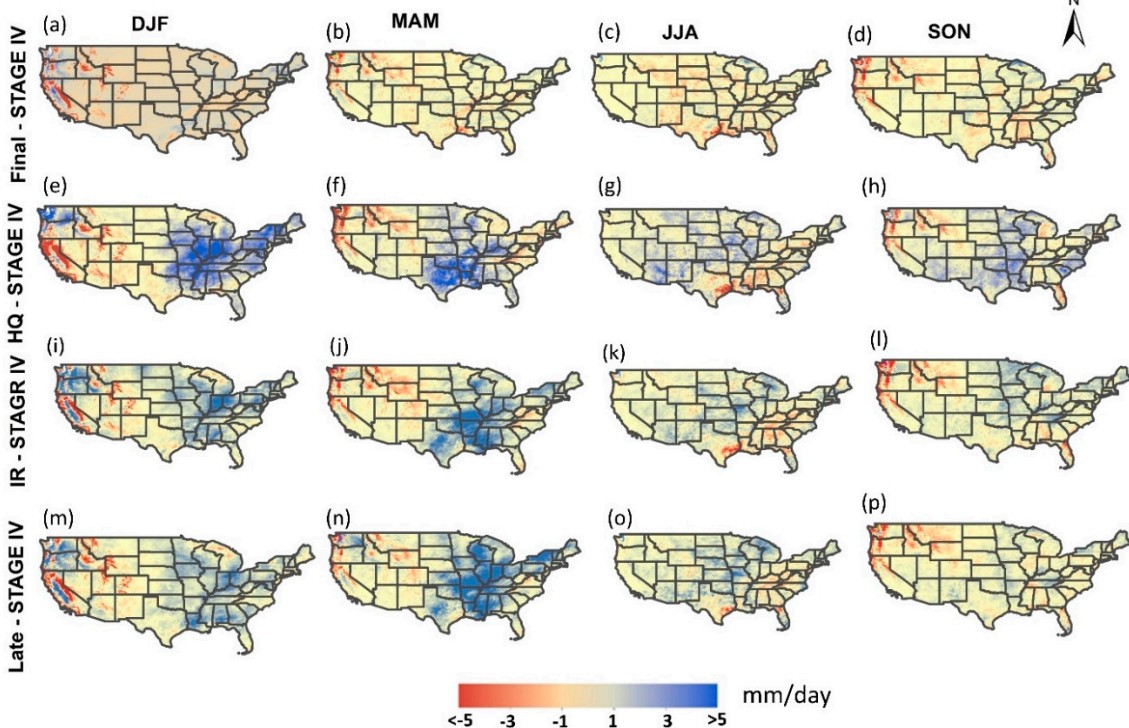

**Figure 2.** Seasonal spatial distribution of the difference in mean daily precipitation between IMERG products and STAGE IV for three years (20152017). Average intensity includes zero precipitation rate. Each row is labeled with IMERG product minus stage IV data. IMERG Final—Stage IV in panel (**a**–**d**), IMERG HQ—Stage IV in panel (**e**–**h**), IMERG IR—Stage IV in panel (**i**–**l**), IMERG Late—Stage IV in panel (**m**–**p**).

Figure 3 shows a precipitation event on 29 December 2017, over the Washington and Oregon states (Figure 3a). This event contains both rain and snowfall, rainfall mainly occurs over the western and southern parts and snowfall occurs over the northeastern part of the study area as can be seen from the liquid probability (Figure 3h) and wet-bulb temperature (Figure 3g) maps. The snow- and ice-covered surfaces are also shown in Figure 3i using the autosnow product. Stage IV precipitation map (Figure 3b) is used as a comparison reference. The white areas in Figure 3b represent regions with no data either because it is over the ocean located in the western part of the region (stage IV has no coverage over the ocean) or missing radar data inland (Figure 3a). As discussed in Section 2, IMERG-Final and IMERG-Late are produced by combining the IR-based and PMW-based precipitation estimates. IMERG-Final bias-adjusts IMERG-Late using rain gauges at a monthly scale. While the bias adjustment is often effective at a monthly scale, it does not necessarily improve the product at a submonthly scale, which could be a reason for the considerable difference between the IMERG-Final and stage IV products. Furthermore, the area includes different climate regions (e.g., based on the Köppen-Geiger climate classification; see Figure S3) and topographic complexity that may affect the retrievals and contribute to the observed differences. IMERG-Late uses IR-based precipitation over snow and ice surfaces and when PMW overpasses are far in time from each other. Therefore, over the northeast, IR-precipitation is directly used in IMERG-Late products as can be inferred from Figure 3d,f. Over the rainfall area, IMERG-Late is produced by combining the IMERG-IR and IMERG-HQ products as can be inferred from Figure 3e,f.

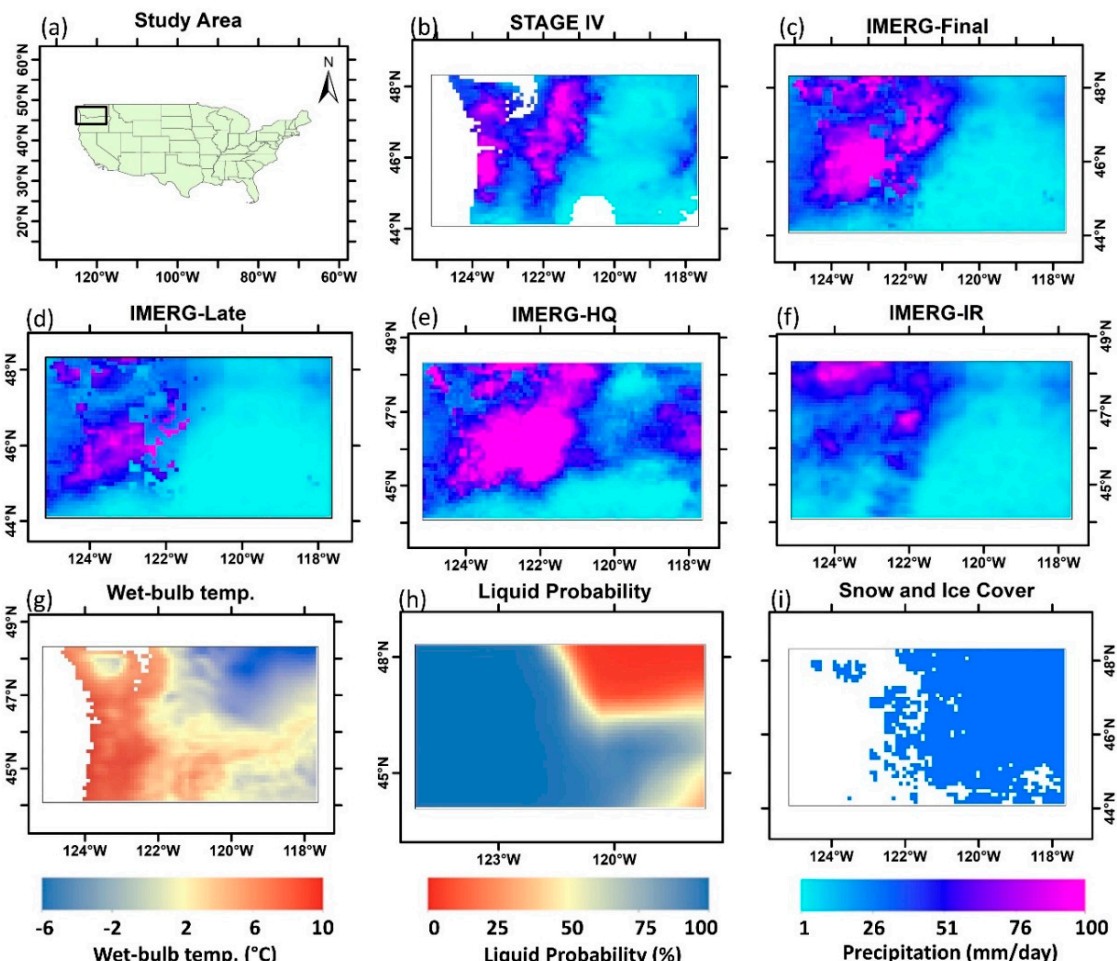

**Figure 3.** A precipitation event on 29 December 2017 for a selected study area shown in (**a**). Precipitation intensity from STAGE IV and IMERG products are shown in panels (**b**–**f**). Wet-bulb temperature, liquid probability, and snow- and ice-covered surfaces are shown in panels (**g**–**i**), respectively.

As discussed in the introduction section, there is a need for a more detailed analysis of the performance of IR (represented by IMERG-IR) and PMW (represented by IMERG-HQ) precipitation estimates over snow and ice surfaces that is investigated in the next section.

*3.2. Assessment of the IMERG Products as a Function of Precipitation Rate, Surface, and Environmental Conditions*

This section compares different IMERG products (e.g., IMERG-Final, IMERG-HQ, IMERG-IR and, IMERG-Late) with the stage IV precipitation estimates by considering precipitation intensity, temperature effects ($T_w$; also related to the precipitation phase), and surface type (i.e., surface with or without snow and ice cover). To investigate how the performance of the precipitation products varies with precipitation rate, five different thresholds (e.g., 0, 0.1, 0.3, 1 and, 2 mm/h) were used and statistical scores were calculated for precipitation events identified by these thresholds (Figures 4 and 5). In other words, precipitation rates smaller than the thresholds were set to nonprecipitating events and precipitation rates equal or greater than the threshold were considered as precipitation events.

Figure 4 shows the performance of different IMERG products over snow- and ice-covered surfaces as a function of $T_w$, plotted in the X-axis. Five different intensity thresholds are used to assess how products perform under events with higher precipitation rates. The top row shows the number of samples used in the analysis (Figure 4a–d), followed by CC, POD, FAR, bias, volume bias (VBias), and HSS in the lower rows. Note that the ideal score for bias and VBias is one as discussed in Section 2. In general, IMERG-Late and IMERG-Final are similar to IMERG-IR, because by design IMERG uses IMERG-IR over snow and ice surfaces. IMERG-Final can be slightly different from IMERG-Late due to the monthly bias adjustment utilized in IMERG-Final. Comparison of IMERG-IR with IMERG-HQ, as a function of $T_w$ and precipitation intensity, enables us to assess whether the use of IR instead of PMW precipitation in IMERG is effective or not, especially at cold temperatures and over snow- and ice-covered regions. Figure 4 shows that as $T_w$ decreases CCs tend to decrease for all the products. While IMERG-HQ is more sensitive than IMERG-IR to changes in $T_w$, IMERG-HQ has higher CC than IMERG-IR at all $T_w$ bins (Figure 4e–h). This is the case for all precipitation intensities, but note that CC is generally lower once we focus on higher precipitation rates.

IMERG-HQ has a higher POD than IMERG-IR at $T_w$ greater than ~ −10 °C, but at colder temperatures POD of IMERG-HQ is generally lower than IMERG-IR (Figure 4i–l). Overall, both IMERG-IR and IMERG-HQ show reduction in POD at colder temperatures, although this is not necessarily the case for precipitation rates greater than 1 mm/h, in which both IMERG-IR and IMERG-HQ show a slight increase in POD as $T_w$ decreases. Note that precipitation is well detected (i.e., POD is about 1) by both products for precipitation intensities greater than 1 mm/h. Figure 4m–p shows that FAR tends to be higher at colder $T_w$ and lower at higher precipitation rates for both IMERG-IR and IMERG-HQ. IMERG-HQ, however, has a slightly lower FAR than IMERG-IR overall. This results in IMERG-HQ having generally higher HSS than IMERG-IR, almost at all $T_w$ ranges, except at very cold temperatures (e.g., $T_w$ < −15 °C) where the two products are comparable (Figure 4z–aa). With respect to bias (Figure 4q–t), both IMERG-IR and IMERG-HQ show values around 1 for precipitation intensities greater than 0.1 mm/h, but at colder temperatures (e.g., $T_w$ less than −10 °C) IMERG-HQ tends to under detect (bias < 1) and IMERG-IR tends to over detect (bias > 1) precipitation occurrences. When precipitation intensities lower than 0.1 mm/h are included in the analysis, both IMERG-IR and IMERG-HQ show larger bias values, suggesting that the products tend to have large false detection of light precipitation as can also be noticed from the FAR plots. This is not necessarily the case for IMERG-HQ for $T_w$ < −10 °C where the products tend to under detect precipitation incidences. VBias plots (Figure 4u–x) show that over snow- and ice-covered surfaces both IMERG-HQ and IMERG-IR tend to overestimate precipitation amount, except for IMERG-HQ for light precipitation at cold temperatures (e.g., $T_w$ less than −12 °C). This is consistent with previous studies showing that PMW tends to underestimate light precipitation over cold surfaces [21]. Clearly, the impact of PMW underestimation is larger in high latitude regions,

where light precipitation is dominant [46,47]. Note that for precipitation rates higher than 1 mm/h, there are large overestimations by both IMERG-HQ and IMERG-IR at colder temperatures that might be related to the confusion of the retrieval methods over snow- and ice-covered surfaces. As can be seen, the bias adjustment employed in IMERG-Final can only slightly improve the IMERG-Late product, likely because the adjustment is performed at a monthly scale, while statistical scores shown in Figure 4 are based on the instantaneous matchups. Overall, results show that IMERG-HQ may outperform IMERG-IR over snow and ice surfaces, at least over the CONUS, suggesting that unconditional use of IR as a replacement for IMERG-HQ over snow and ice surfaces needs to be revisited.

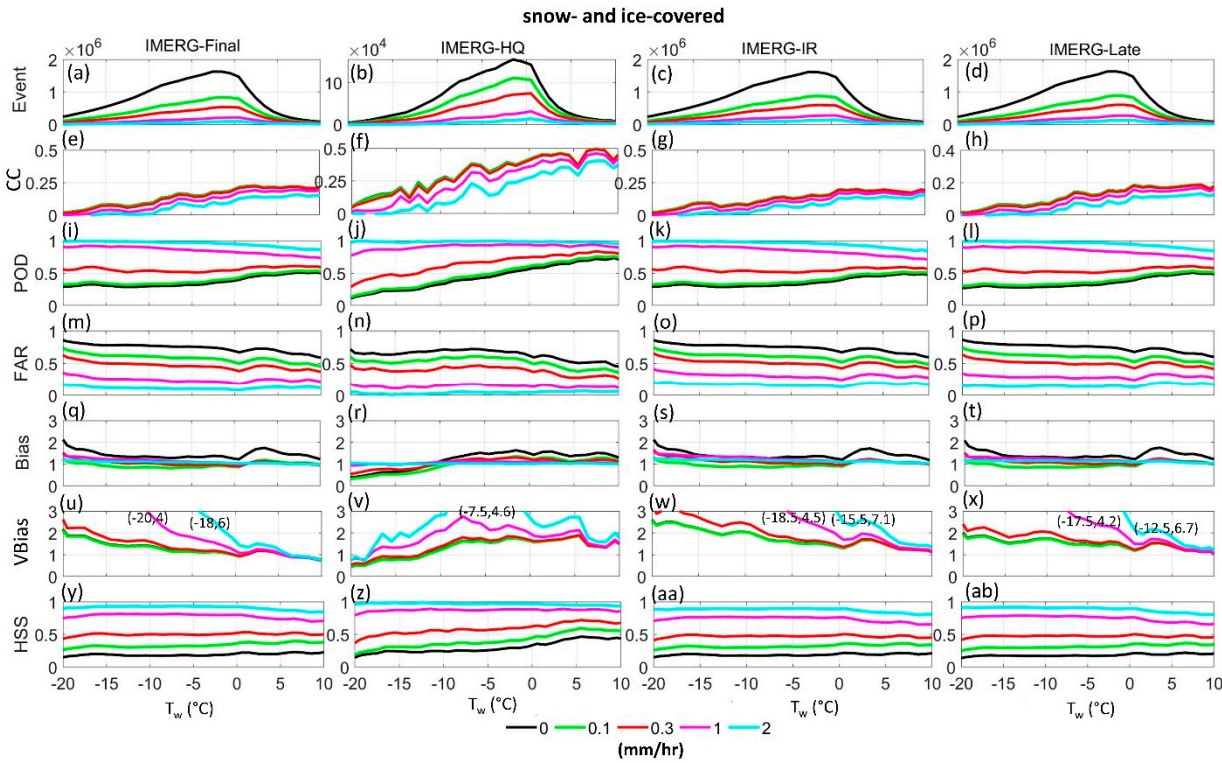

**Figure 4.** Comparison of different IMERG products (from left to right column: final, HQ, IR and, late) with stage IV data for three years (2015–2017) over CONUS over snow- and ice-covered surfaces. Orographic effects are excluded using mountains mask. The X-axis indicates wet-bulb temperature 1 °C bins. Each color indicates the threshold for rainfall intensity from 0 mm/h (in black) to 2 mm/h (in cyan). The number of events for each threshold is shown in the first row (**a–d**). The second row to the seventh row shows statistical indices for correlation coefficient (CC), probability of distribution (POD), false alarm ratio (FAR), bias, volume bias (VBias) and, Heidke skill score (HSS) shown in panels (**e–ab**).

Figure 5 is similar to Figure 4, but it is over snow- and ice-free surfaces. Here IMERG-Late and IMERG-Final mainly follow IMERG-HQ scores rather than IMERG-IR. However, the impact of IMERG-IR on IMERG-Late (thus IMERG-Final) can be seen (at least in the CC plot), as IR is still used in IMERG-Late when the time-distance between successive PMW overpasses is large. It can be seen that both IR and PMW products tend to have better scores over snow- and ice-free than over snow-and ice-covered surfaces and both show worse skill scores as $T_w$ decreases. Furthermore, IMERG-HQ tends to outperform IMERG-IR almost regardless of the $T_w$ and intensity ranges. Both products also show higher skill scores when light precipitation is removed from the analysis, except for CC and VBias. Both IMERG-IR and IMERG-HQ show large VBias when precipitation intensities greater than 1 mm/h are assessed at lower temperatures, but VBias is clearly larger for IR at $T_w$ less than 5 °C. At $T_w$ less than 0 °C, IMERG-HQ shows slight underestimation for events that include light precipitation. Overall, it appears that the use of IMERG-IR performs worse than IMERG-HQ over snow- and ice-free surfaces. Note that IMERG-HQ

does not include morphed PMW estimates, so the outcomes of this analysis do not apply to the morphed PMW estimates that may show less skill at a longer time distance from the time of PMW observations [48].

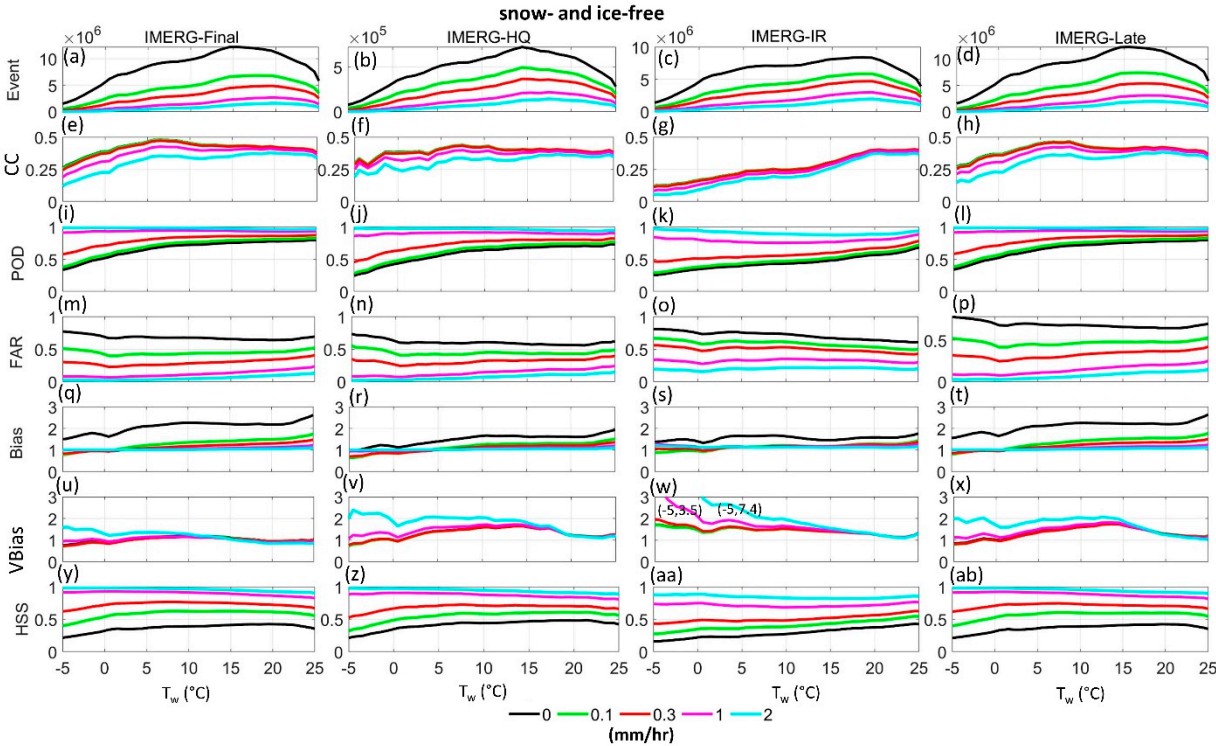

**Figure 5.** Similar to Figure 4, but for snow- and ice-free surfaces.

Figure 6 facilitates comparing IMERG-IR with IMERG-HQ as a function of $T_w$ over snow- and ice-covered (left-column) and snow- and ice-free (right-column) surfaces. Here, a fixed threshold of 0.3 mm/h is used for precipitation delineation. It can be seen that IMERG-HQ outperforms IMERG-IR over snow- and ice-free surfaces across all temperatures, although it shows slightly worse bias and VBias than IMERG-IR for $T_w < \sim 5 \,^\circ$C (Figure 6h) and $T_w$ between 10 $^\circ$C and 20 $^\circ$C (Figure 6j), respectively. Over snow- and ice-covered surfaces IMERG-HQ outperforms IMERG-IR in terms of CC (Figure 6a) and FAR (Figure 6e), regardless of $T_w$. However, IMERG-IR shows higher POD and HSS than IMERG-HQ for $T_w < \sim -12 \,^\circ$C (Figure 6c,k), but this is along with higher FAR and bias (Figure 6e,g). Figure 6i shows a significantly large VBias for IMERG-IR, especially for $T_w < -8 \,^\circ$C. Other indices can also be used for comparison. Figure S4 compares the IMERG products using the Kling–Gupta efficiency coefficient (KGC) [49] which is a quantitative score and considers the distance between mean and variance of observed and estimated time series and their correlation. As can be seen, the overall conclusion from KGC plots is that IMERG-HQ outperforms IMERG-IR for most $T_w$ bins, especially for colder bins regardless of the surface type. This suggests that replacement of precipitation estimate from PMW with IR over snow and ice surfaces may not be effective. The analysis can also be extended by calculating systematic error (SE) and random error (RE) for each IMERG product over two surface covers and for different $T_w$ ranges. SE and RE are absolute mean relative error and normalized root mean square error, respectively, and are described in the supplementary file as well as in [50]. Table S1 suggests that over snow- and ice-covered surfaces IMERG HQ has smaller RE than SE over all $T_w$ ranges. In contrast, IMERG-IR shows larger RE than SE over all $T_w$ ranges. As reduction of SE is easier than RE [51], efforts such as bias correction may further improve the PMW precipitation estimates.

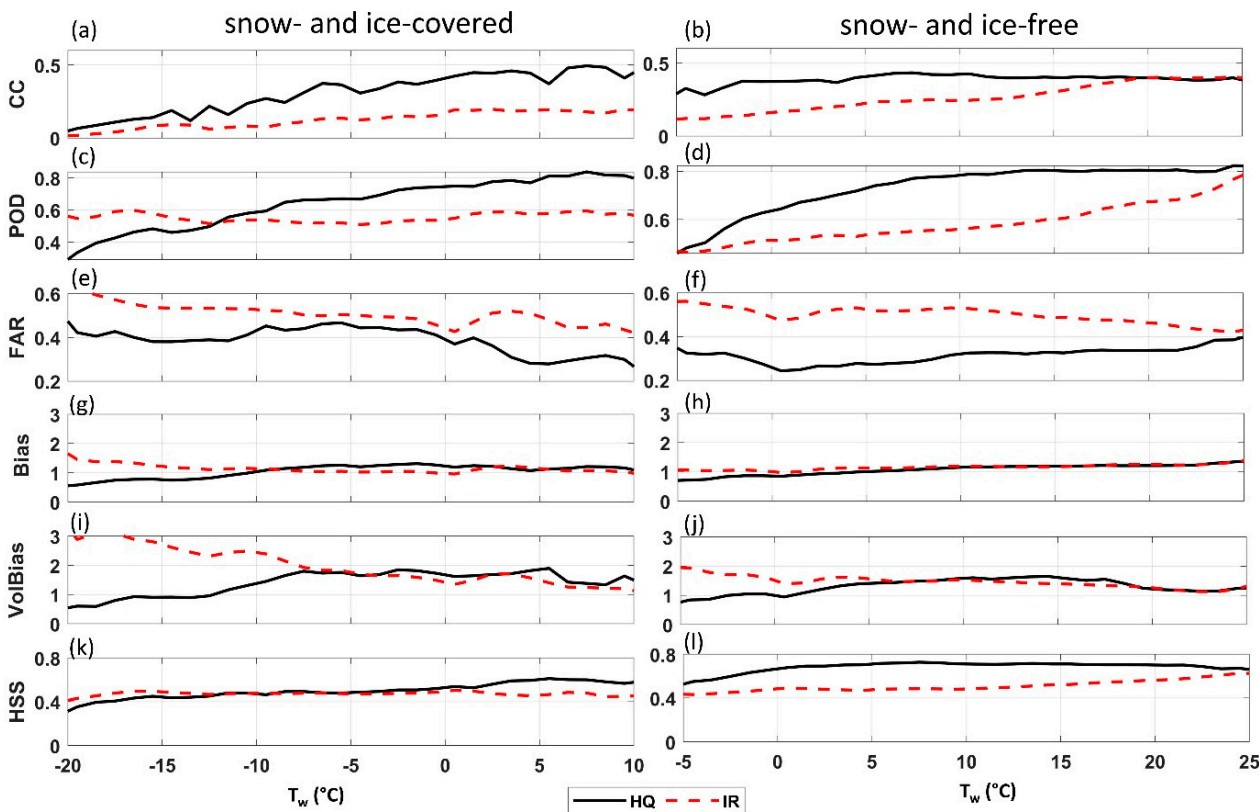

**Figure 6.** Comparison of IMERG-HQ and IMERG-IR over snow- and ice-covered and snow- and ice-free surfaces using three years (2015–2017) of data over CONUS. Stage IV is used as a reference. Dashed and solid lines represent IMERG-IR and IMERG-HQ in panels (**a**–**l**), respectively. Precipitation is delineated from no precipitation using a threshold of 0.3 mm/h. Orographic effects are excluded using mountains mask.

### 3.3. Performance of Individual PMW Precipitation Estimates

In the previous section precipitation estimates from a combination of PMW sensors (i.e., through IMERG-HQ) were compared with precipitation estimates from geostationary IR observations. However, IMERG-HQ is composed of precipitation estimates from several PMW sensors that are identified in the IMERG output fields. Here the performance of individual PMWs is compared with IMERG-IR as a function of $T_w$ and using stage IV as reference. Similar to the previous section, analysis is conducted separately on snow- and ice-covered and snow- and ice-free surfaces. Timespan, study area, and procedure are the same as in the last section and a threshold of 0.3 mm/h is used to delineate precipitation from nonprecipitation. Orographic effects are excluded using mountains mask. Note that the analysis is based on sensor type, not satellite, so if a sensor is available on more than one platform a combination of the sensors is used. Furthermore, IMERG prefers PMW imagers over PMW sounders, so if they coincide, precipitation estimates from PMW imagers are used in IMERG-HQ (Huffman et al. 2020). Based on Figure 7, and over snow- and ice-covered surfaces, the following observations are highlighted: (1) all PMW sensors, except AMSR-2, have better CC than IMERG-IR regardless of $T_w$ (Figure 7c), and AMSR-2 shows the lowest CC among all the studied products for $T_w < \sim-5\,°C$, (2) SSMIS shows the best and AMSR-2 shows the worst POD among all the PMW sensors, IMERG-IR has better POD than SSMIS for $T_w < -10\,°C$, better POD than GMI, MHS and ATMS for $T_w < -3\,°C$, and better POD than AMSR-2 for $T_w < 5\,°C$ (Figure 7e).

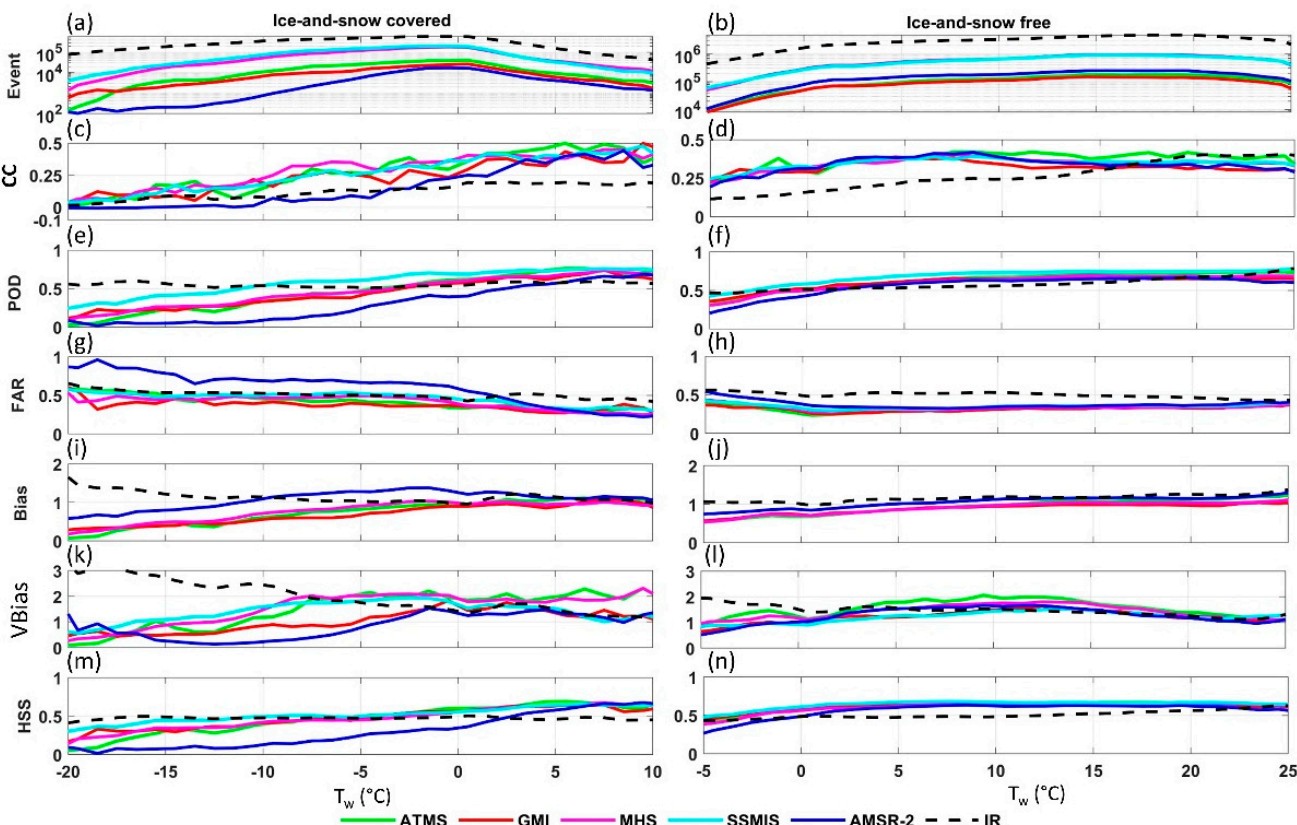

**Figure 7.** Comparison of precipitation estimates from different passive microwave sensors (used in IMERG-HQ) with IMERG-IR using stage IV data as a reference. Three years (2015–2017) of data over CONUS are used. Orographic effects are excluded using mountains mask. Event numbers is shown in panels (**a**,**b**) and coefficients in panels (**c**–**n**).

POD of AMSR-2 is almost zero for $T_w < -10\ ^\circ$C, (3) for FAR (Figure 7g), IMERG-IR has higher FAR than all PMW sensors except AMSR-2 that shows higher FAR than IMERG-IR for $T_w < 2\ ^\circ$C or generally snowfall, (4) for bias, PMW and IMERG-IR show good skill (i.e., bias near 1) for $T_w > 0\ ^\circ$C, but for $T_w < 0\ ^\circ$C, IMERG-IR tends to show bias values greater than one and PMW sensors tend to show bias values smaller than 1. The biases get worse as $T_w$ decreases. AMSR-2 has Bias values greater than 1 for $T_w > -10\ ^\circ$C, but bias values less than one for $T_w < -10\ ^\circ$C, (5) in terms of volume bias, individual sensors tend to perform differently, but all of them overestimate stage IV precipitation amount for $T_w > -15\ ^\circ$C, except GMI and AMSR-2 that tend to underestimate for $T_w < \sim -5\ ^\circ$C. The overestimation of IMERG-IR is more significant than all PMW sensors at $T_w < -8\ ^\circ$C at which AMSR-2′s VBias reaches almost zero. The increase of AMSR-2 VBias at $T_w < -15$ could be due to unstable sampling (note that AMSR-2 has the lowest sampling among all other sensors; Figure 7a), (6) as an overall score for precipitation detection, HSS suggests that IR is superior to the individual PMW sensors for $T_w < -10\ ^\circ$C (2 $^\circ$C for AMSR-2), although SSMIS shows lower HSS than IMERG-IR only for $T_w < -15\ ^\circ$C, AMSR-2 show almost no skill at $T_w < -15\ ^\circ$C. One reason for the relatively poor performance of AMSR-2 compared to other PMW products could be the lack of high frequency (e.g., greater than 90 GHz) and also sounding channels in AMSR-2 compared to other PMW sensors. It has been shown that high-frequency MW and sounding channels are valuable in retrieving snowfall and can be used to mask out scattering signals from snow- and ice-covered surfaces that might otherwise interfere with scattering signals from ice particles in clouds (e.g., Skofronick-Jackson, Kulie [3]).

PMW sensors perform generally better over snow- and ice-free than over snow- and ice-covered surfaces. PMW sensors have higher CC than IMERG-IR for $T_w < 15\ ^\circ$C, but at warmer $T_w$ they are relatively comparable (Figure 7d). Over snow- and ice-free surfaces,

PMW sensors tend to have higher POD than IMERG IR (except for $T_w < \sim 0$ °C at which PMW sensors' POD decreases with reduction in $T_w$; Figure 7f), IMERG-IR shows higher FAR than PMW sensors for all temperatures (Figure 7h), and both bias and VBias of PMW sensors and IMERG-IR are fairly comparable for $T_w > 0$ °C. At $T_w < 0$ °C, IMERG-IR tends to show a slight overestimation in Bias (Figure 7j) and large overestimation in VBias (Figure 7l), but PMW sensors tend to underestimate both precipitation occurrence (bias) and amount (VBias) at colder temperatures. As an overall metric, HSS suggests that IMERG-IR tends to outperform PMW sensors at $T_w < 0$ °C for AMSR-2 and $T_w < -4$ °C for other sensors over snow- and ice-free surfaces.

## 4. Concluding Remarks

Accurate estimation of precipitation is important for water cycle studies and various hydrologic applications. A long-standing challenge for remote sensing products has been an estimation of precipitation in cold regions, especially over snow and ice surfaces. Here, using three years of Stage IV data (2015–2017) over the CONUS, the performance of various products of IMERG is investigated as a function of near-surface wet-bulb temperature (used for precipitation phase detection), precipitation intensity, and surface type (i.e., with and without snow and ice on the surface). The IMERG products include precipitation estimations from infrared (IR), combined passive microwave (PMW) sensors, and a combination of the precipitation estimate from IR and PMW sensors, that are either bias adjusted using in situ data (IMERG-Final) or not (IMERG-Late). In the analysis, steep mountainous regions were eliminated to reduce complexities that might be due to the orographic enhancement of precipitation.

Results show that precipitation estimates from PMW products generally have better statistics than IR over snow- and ice-free surfaces. Over snow- and ice-covered surfaces, PMW products (except AMSR-2) show a higher correlation coefficient (with stage IV data) than IR. Both IR and PMW precipitation products tend to overestimate precipitation over snow and ice surfaces, but, at colder temperatures (e.g., $T_w < -10$ °C), PMW products tend to underestimate while IR product continues to show large overestimations. With respect to precipitation occurrence, both PMW and IR products show considerably higher skill in capturing intense precipitation than light precipitation rates. PMW sensors outperform IR over snow- and ice-free surfaces and also show higher overall skill in detection precipitation occurrence over snow- and ice-covered regions, but not necessarily at $T_w$ colder than $-5$ °C. Generally, AMSR-2 performs worst and SMMIS performs best among the studied PMW sensors that also include GMI, MHS, and ATMS. The results suggest that the current approach of IMERG, replacing PMW with IR precipitation estimates over snow- and ice-covered surfaces, needs further investigations and might need to be revised.

Previous studies show poor performance of PMW-based precipitation in cold regions, especially in the presence of the snow or ice on the surface. This study confirms this issue. It also indicates that the majority of PMW sensors could outperform IR-based precipitation in cold temperatures over snow- and ice-covered surfaces. It should be mentioned that assessment of PMW- and IR-based precipitation in higher latitudes might be different from that over CONUS. This could be due to the differences in environment and type of precipitation (e.g., atmosphere is generally drier and light precipitation is more frequent in high latitudes than CONUS). Future studies can include other variables such as total precipitable water to account for the effect of dry and moist environment in the analysis. Furthermore, for regions poleward of 60° S/N, PERSIANN-CCS is not available (i.e., due to the low quality of geostationary IR images), thus precipitation estimates from other IR sensors such as the atmospheric infrared sounder (AIRS) [41] or the advanced very-high-resolution radiometer (AVHRR) [5] could be considered. AIRS is used in the Global Precipitation Climatology Project (GPCP) product [42] in high latitudes. In the meantime, new algorithms for precipitation retrieval from PMW is being developed that may outperform the current estimates (e.g., [3,43,44]). Future studies are needed to inter-

compare these products, so the best product from PMW or IR can be used in IMERG and other multisensor products.

**Supplementary Materials:** The following are available online at https://www.mdpi.com/article/10.3390/rs13142726/s1, Figure S1: Mask of high and high-scatter mountains based on K3 mountain maps. Figure S2: Annual average precipitation for three years (2015–2017) over CONUS for Stage IV (a), IMERG-Final (b), IMERG-HQ (c) and, IMERG-Late(d). Figure S3: Revised Köppen-Geiger climate classification maps at 1-km resolution for present condition (1980–2016) by Beck et al. 2018. For definition of each classification in the legend see Beck et al. 2018 [52]. Figure S4: Comparison of IMERG products using KGC over snow- and ice-covered (top panel) and snow- and ice-free (bottom panel) surfaces using three years (2015–2017) of data over CONUS. Stage IV is used as a reference. Table S1: Percent systematic error (SE) and random error (RE) for different IMERG products for different Tw ranges and surface types.

**Author Contributions:** Conceptualization, A.A. and A.B.; methodology, A.A.; software, A.A.; validation, A.B., A.A.; formal analysis, A.A.; investigation, A.A.; resources, A.B.; data curation, A.A.; writing—original draft preparation, A.A.; writing—review and editing, A.B.; visualization, A.A.; supervision, A.B.; project administration, A.B.; funding acquisition, A.B. Both authors have read and agreed to the published version of the manuscript.

**Funding:** Financial support was made available from NASA MEaSUREs (NNH17ZDA001N-MEASURES) and NASA Weather and Atmospheric Dynamics (NNH19ZDA001N-ATDM) grants.

**Data Availability Statement:** The NCEP-STAGE-IV product was obtained from Earth Observing Laboratory (EOL) website at https://data.eol.ucar.edu/ (accessed on 1 June 2020). IMERG products are available from the Goddard Earth Sciences Data and Information Services Center (GES DISC https://disc.gsfc.nasa.gov/ERA5 (accessed on 1 July 2020) data is downloaded from the Climate Data Store (https://cds.climate.copernicus.eu/#!/home (accessed on 1 July 2020)) and K3 mountain maps obtained from USGS website at https://rmgsc.cr.usgs.gov/outgoing/ecosystems/Global/ (accessed on 1 January 2021).

**Conflicts of Interest:** The authors declare no conflict of interest.

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
