# Peer review of "Investigating Various Products of IMERG for Precipitation Retrieval over Surfaces with and without Snow and Ice Cover"

_remotesensing, doi:10.3390/rs13142726_

Round 1

Reviewer 1 Report

I think the authors carried out a large amount of work to evaluate IMERG precipitation product over CONUS. Authors need to present a detailed comprehensive introduction section along with detail error analysis results. I recommend the manuscript for publication after the following major changes:

  • Evaluating the performance of IMERG version 6 products is also crucial in different countries specifically subtropical/tropical climate countries such as Brazil, Peru, Colombia as well as Asia. Moreover, IMERG showed poor performance over Northern China (Chen 2016). Similarly, Murali et al. 2017 showed underestimation for the IMERG products compared to gauge precipitation over the Indian subcontinent. Other studies concluded that the performance of IMERG is relatively unsatisfactory for the seasonal variation for certain tropical/subtropical regions like Ganges Brahmaputra Meghna (GBM) basin is paramount (Bhuiyan, et al. 2020). Such inconsistencies in IMERG products might be associated with various sources of errors causing a detrimental impact on the hydrologic investigation. Please introduce these works and their potential impact. The authors should explain this aspect in the introduction section. Otherwise, the readers cannot see the importance of your study over other previous studies. Also, you need to provide more IMERG related literature reviews in the introduction section associated with the research gap/limitation. I am providing a few, but you will get more literature related to IMERG evaluation and error correction.

Chen, F.; Li, X. Evaluation of IMERG and TRMM 3B43 Monthly Precipitation Products over Mainland China. Remote Sens. 2016, 8, 472.

Murali Krishna, U.V.; Das, S.K.; Deshpande, S.M.; Doiphode, S.L.; Pandithurai, G. The assessment of Global Precipitation Measurement estimates over the Indian subcontinent. Earth Space Sci. 2017, 4, 540–553.

Bhuiyan, et al. 2020: Machine Learning-Based Error Modeling to Improve GPM IMERG Precipitation Product over the Brahmaputra River Basin. Forecasting 2020, 2, 248-266.

Anjum, Muhammad Naveed, et al.2019 "Assessment of IMERG-V06 precipitation product over different hydro-climatic regimes in the Tianshan Mountains, North-Western China." Remote Sensing 11.19 (2019): 2314 (2)

Moazam, et al. 2020"A Comprehensive Evaluation of GPM-IMERG V06 and MRMS with Hourly Ground-Based Precipitation Observations across Canada”. Journal of Hydrology. 2020 Dec 29:125929.

  • As IMERG showed unsatisfactory performance over few subtropical/tropical countries. Why did you choose to use the IMERG product instead of the PERSIANN, CMORPH, GSMAP, or MSWEP?
  • Why did you using Stage-IV product as reference precipitation over high-resolution multiple-radar/multisensor (MRMS) precipitation or gauge observation?.
  • Your study area has a high dependency on local climate, topographic complexity. Can you provide detail climatic information for the selected study areas? You should provide a show Köppen–Geiger climatic zones on the map.
  • Can you provide a table for error analysis in terms of systematic error (absolute mean relative error) and random error(normalized root mean square error) for the evaluation of different precipitation products?
  • you introduced several Statistical metrics and Categorical skill scores. Also, can you provide the Kling Gupta coefficient test for the efficient test?
  • Past research showed that during summer when convective precipitation events occur, more severe underestimations were found in reanalysis precipitation. In the discussion section to verify IMERG, you should compare your results with the previously achieved results. Also discussed these issues in the limitation section.

Reviewer 2 Report

Line no. 34: Provide acronym for IR.

Line 44: Add a sentence or two describing how the blending of the PMW and IR is done in PERSIANN – CCS.

Line 154: Also provide the resolution in km or m. Stopped at line 180.

What about the precipitation pattern, annual average during the 3-years of study period, 2015-2017?

What is the reason to select Washington and Oregon? Are there any other days within the study period when both rain and snow falls?

Line 296: Typo “IMER-IR”.

To understand the performance of the IMERG product over snow or ice, figure 4 should be a comparison using a IMERG product resolution, i.e., 0.1x0.1 degree not 1 x 1 degree bin.

Reviewer 3 Report

Dear Authors,

All my comments are included in the .pdf version.

Best regards,

Reviewer 4 Report

Reliable documentation of precipitation is essential for practical applications as well as for monotoring effects of climate change. The manuscript takes a critical look at satellite-based precipitation data for the US. In particular it evaluates strengths, weaknesses and differences betewen infrared (IR) and passive microwave (PMW). The authors argue that PMW should not be easily abandonned, as the performance of IR-based data may not be as good as orignally thought.

The authors describe the challenge, their approach and the results in a clear and well-written way. Figures are of high quality. I find the Discussion section too brief, in particular as the authors criticize an IMERG procedure. The Discussion section would be a good place to provide more detail on the proposal, including a suggested way forward.

The Conclusions are again well-written.

Whilst the abstract is technically ok and contains all the key information, it might benefit from some simplification to make it more readable. I am thinking mostly of non-specialists who ae confronted with a large number of acronyms and abbreviations. Can the abstract be slightly modified to make it more reader-friendly?

In general, this paper will add important data and perspective to the debate. It is clear that there are no perfect solutions, but by comparing and discussing options, workflows can be further refined and biases reduced. I therefore consider the manuscript a useful and timely contribution.

Minor points:

Define CONUS (Contiguous United States) for readers outside the US

Line 445: Although performance

Line 446-447: “have not been fully discovered”. Better: Evaluated, studied?

Line 449: it also indicates that the majority

Round 2

Reviewer 1 Report

The authors significantly improved the quality of the paper by addressing most of the comments. This research work will be very effective in the water resources community. I recommend the manuscript for publication. Congratulation to the authors!!

Author Response

Thank you for your constructive comments. They helped us to make our paper better.

Reviewer 3 Report

Dear authors,
The manuscript has been improved in the current version. However, it needs minor revision:
Page 1, line 22: Avoid repetition of words used in the title;
Page 1, line 36: PMW has been already described in the abstract;
Page 1, line 103: Describe how your research can help decision-makers in different scientific areas;
Page 5, line 221: It is necessary to create a section of discussion and separate it from the results section;
Page 14, line 473: Conclusions must be clearer and concise, answering the main goals of your study.
